# Enteropathogenic *Yersinia* with Public Health Relevance Found in Dogs and Cats in Finland

**DOI:** 10.3390/pathogens13010054

**Published:** 2024-01-05

**Authors:** Maria Fredriksson-Ahomaa, Thomas Grönthal, Viivi Heljanko, Venla Johansson, Merja Rantala, Annamari Heikinheimo, Riikka Laukkanen-Ninios

**Affiliations:** 1Department of Food Hygiene and Environmental Hygiene, Faculty of Veterinary Medicine, University of Helsinki, P.O. Box 66, 00014 Helsinki, Finland; viivi.heljanko@helsinki.fi (V.H.); venla.johansson@helsinki.fi (V.J.); annamari.heikinheimo@helsinki.fi (A.H.); riikka.laukkanen-ninios@helsinki.fi (R.L.-N.); 2Animal Health Diagnostic Unit, Finnish Food Authority, Mustialankatu 3, 00790 Helsinki, Finland; thomas.gronthal@ruokavirasto.fi; 3Department of Equine and Small Animal Medicine, Faculty of Veterinary Medicine, P.O. Box 57, 00014 Helsinki, Finland; merja.rantala@gmail.com; 4Microbiology Unit, Finnish Food Authority, Mustialankatu 3, 00790 Helsinki, Finland

**Keywords:** *Yersinia enterocolitica*, *Yersinia pseudotuberculosis*, pets, virulence, resistance, characteristics

## Abstract

Yersiniosis is a common zoonotic enteric disease among humans, which has been linked to pigs and contaminated food, especially pork. The epidemiology of yersiniosis is still obscure, and studies on yersiniosis in pets are very scarce. In this study, we performed pheno- and genotypic characterisation of 50 *Yersinia* strains isolated from pets in Finland between 2012 and 2023. *Y. enterocolitica* 4/O:3/ST135, the most common type in human yersiniosis, was also the most common type (68%) found in clinical faecal samples in our study. Also, human pathogenic *Y. enterocolitica* 2/O:9/ST139 and *Y. pseudotuberculosis* O:1/ST9 and O:1/ST42 strains carrying all essential pathogenic genes were identified. Three *Y. enterocolitica* 4/O:3/ST9 strains were multi-drug-resistant and two of them were highly related, showing one allelic difference (AD) with core genome multi-locus sequence typing. Non-pathogenic, genotypically highly diverse *Y. enterocolitica* 1A strains, showing more than 1000 ADs and missing the essential virulence genes, were also recognised in dogs and cats. Our study demonstrates that pets can excrete human pathogenic *Yersinia* in their faeces and may serve as an infection source for human yersiniosis, especially in families with small children in close contact with their pets.

## 1. Introduction

*Yersinia* are Gram-negative rods quite recently reclassified into the *Yersiniaceae* family within the *Enterobacterales* order [1]. Two species, *Yersinia enterocolitica* and *Y. pseudotuberculosis*, are enteropathogens causing enteric yersiniosis in humans and animals [2]. Yersiniosis was the third most reported enteric disease in 2022 within the EU, and the highest detection rate was reported in under-five-year-old children [3]. The highest notification rates were reported in Denmark (12.7/100,000 inhabitants) and Finland (7.4/100,000). The infections were primarily caused by *Y. enterocolitica* and only rarely by *Y. pseudotuberculosis*. Symptoms range from mild diarrhoea to systemic infection. Typically, the infection is an uncomplicated enteric disease with diarrhoea and abdominal pain, and only in some cases do extraintestinal complications occur [2].

*Y. enterocolitica* is divided into six biotypes (1A, 1B, 2–5) based on biochemical characteristics and into numerous serotypes based on its O antigen. However, only certain biotype and serotype combinations have been associated with human and animal infections. The most common type reported in the EU is bioserotype 4/O:3, followed by 2/O:9 [3]. Biotype 1A strains, regarded as non-pathogenic, are very commonly found, especially in non-human samples. They are missing the virulence plasmid (pYV) and essential chromosomal genes needed for pathogenesis. Correctly identified *Y. pseudotuberculosis* is regarded as pathogenic, carrying virulence genes both on the pYV plasmid and in the chromosome. *Y. enterocolitica* and *Y. pseudotuberculosis* strains found in the EU are usually sensitive to most antimicrobials used for human and animal treatment [4,5]. However, *Y. enterocolitica* has intrinsic resistance to ampicillin due to the *bla*A gene in its chromosome. 

The epidemiology of enteric yersiniosis is still poorly understood. Several possible transmission routes occur for *Y. enterocolitica* and *Y. pseudotuberculosis.* Occupational contact with pigs and pork consumption are significantly associated with sporadic *Y. enterocolitica* infections [6,7]. The consumption of raw or undercooked pork and drinking untreated water are particularly important risk factors. Dogs and cats are potential sources of human yersiniosis through close contact, especially for young children [8]. Pathogenic *Y. enterocolitica* may be transmitted to dogs and cats via raw pork [9]. However, yersiniosis in dogs and cats has seldom been reported. In an earlier study conducted in Germany, *Y. enterocolitica* was isolated from 4.6% and 0.3% of dog and cat faecal samples, respectively [10].

To obtain more information concerning *Yersinia* strains isolated from clinical samples of dogs and cats in Finland, we pheno- and genotypically characterised the strains. We studied the presence of various virulence and resistance genes and the sensitivity to clinically important antimicrobials among pet *Yersinia* strains to attain more information on their possible relevance to public health. We additionally studied the genetic relationship between the strains using core genome multi-locus sequence typing (cgMLST) to obtain new information on genotypes circulating among dogs and cats.

## 2. Materials and Methods

### 2.1. Yersinia enterocolitica and Yersinia pseudotuberculosis Strains

We characterised 50 *Yersinia* strains isolated from pets between 2012 and 2023 at the Veterinary Teaching Hospital of the University of Helsinki (Figure 1). Most (92%, 46/50) strains were from dogs and the rest (8%, 4/50) from cats. In total, 38 (76%) *Y. enterocolitica* strains (35 from dogs and 3 from cats) and 12 (24%) *Y. pseudotuberculosis* strains (11 from dogs and 1 from a cat) were pheno- and genotypically characterised. They were isolated from clinical specimens, mostly (88%, 44/50) from the faeces of animals. Six strains were isolated from extra-intestinal sites: from an abscess (n = 2), urine (n = 2), blood (n = 1), and a wound (n = 1).

### 2.2. Phenotypic Characterisation of Yersinia Strains

*Yersinia* strains isolated between 2012 and 2017 were identified using API 20E (BioMerieux, Marcy-l’Etoile, France) and, from 2018 onwards, with MALDI-TOF MS. In total, 38 *Y. enterocolitica* strains were biotyped based on tween esterase activity, indole production, and xylose and trehalose fermentation. All 50 strains were serotyped with commercial antisera (Sifin, Berlin, Germany): 38 *Y. enterocolitica* strains against O:3 and O:9 and 12 *Y. pseudotuberculosis* strains against O:1. The phenotypical susceptibility to antimicrobials was studied with a minimum inhibitory concentration (MIC) method (Senititre, EUVSEC3, Thermo Fisher Diagnostic, Vantaa, Finland), which included 15 antimicrobial agents: amikacin, ampicillin, azithromycin, cefotaxime, ceftazidime, chloramphenicol, ciprofloxacin, colistin, gentamicin, meropenem, nalidixic acid, sulfamethoxazole, tetracycline, tigecycline, and trimethoprim. The plates were incubated at 30 °C for 24 h and the MIC values were read with a Sensititre Vizion instrument (Thermo Fisher Diagnostic). Susceptibility thresholds were interpreted in accordance with EUCAST (https://www.eucast.org/fileadmin/src/media/PDFs/EUCAST_files/Breakpoint_tables/v_13.1_Breakpoint_Tables.pdf, accessed on 1 November 2023).

### 2.3. Genotypic Characterisation of Yersinia Strains

The identification and pathogenicity of the *Yersinia* strains were confirmed with PCR (Table 1). For whole-genome sequencing (WGS), the DNA was extracted using a QIAcube Connect instrument (Qiagen, Hilden, Germany) with a DNeasy Blood and Tissue Kit (Qiagen, Valencia, CA, USA). *Yersinia* cells for DNA extraction were harvested from 2 mL of tryptic soya broth (Oxoid, Basingstoke, UK) after overnight incubation for 20–22 h at 30 °C. DNA quality (A260/A280 ratio) was measured with a NanoDrop ND-1000 spectrophotometer (Thermo Fisher Scientific, Waltham, MA, USA), and DNA quantity with a Qubit 2.0 fluorometer (Invitrogen, Life Technologies, Carlsbad, CA, USA). WGS was performed using an Illumina NovaSeq 6000 (Novogene, Cambridge, UK) with a 2 × 150 bp read length and a targeted genomic coverage of 100×. The reads were assembled de novo with a Unicycler v0.4.8 assembler available on the PATRIC 3.6.12 platform (https://www.bv-brc.org/app/Assembly, accessed on 1 December 2023). The sequence types (STs) were defined with 7-gene MLST based on the Achtman scheme [11] and the core genome sequence types (CTs) were defined using the CGE (Center for Genomic Epidemiology) platform (https://cge.food.dtu.dk/services/MLST/, accessed on 1 December 2023). Ad hoc cgMLST targeting 3839 genes of *Y. enterocolitica* and 2886 genes of *Y. pseudotuberculosis* was performed with Ridom SeqSphere+ software v7.7.5 (Ridom GmbH, Muenster, Germany). A minimum spanning tree (MST), representing pairwise allele distances ignoring missing values of 26 *Y. enterocolitica* 4/O:3 strains, was used to visualise ADs between the strains. Antimicrobial resistance genes were confirmed with Ridom software and ResFinder v4.4.1 on the CGE platform.

## 3. Results

Phenotypic species identification was based on API20E or MALDI-TOF MS. The identification rate (ID%) of API 20E varied between 88.4% and 99.8% for *Y. enterocolitica* and between 98.6% and 99.9% for *Y. pseudotuberculosis*. The identification scores of MALDI-TOF MS varied from 2.00 to 2.44 for *Y. enterocolitica* and from 2.00 to 2.40 for *Y. pseudotuberculosis*, respectively. All strains had a score over 2.00, indicating probable species identification. Most of the *Y. enterocolitica* strains (26/38, 68%) belonged to bioserotype 4/O:3, and four (11%) strains belonged to bioserotype 2/O:9; both types are regarded as pathogenic. Non-pathogenic biotype 1A strains were identified in eight (21%) cases (Table 2). All 12 *Y. pseudotuberculosis* strains belonged to serotype O:1. In six (12%) animals, the infection was extra-intestinal. *Y. pseudotuberculosis* O:1 was identified in five of the six extra-intestinal infections and *Y. enterocolitica* 1A in one abscess from a dog (Table 2).

All *Y. enterocolitica* strains belonging to the bioserotypes 4/O:3 and 2/O:9 carried the chromosomal *ail*, *inv*, and *yst*A genes, and most (97%, 29/30) of the strains also carried the genes *vir*F and *yad*A located on the virulence plasmid pYV (Table 3). All 4/O:3 strains belonged to the sequence type ST135, and all 2/O:9 strains to ST139. One 1A strain carried the *ail* gene associated with virulence. The *yst*B gene, which is often found in non-pathogenic *Yersinia* strains, was detected in five 1A strains; however, *yst*A associated with *Yersinia* pathogenicity was not detected. All eight 1A strains were also negative for the *vir*F and *yad*A genes located on the pYV plasmid. Each also belonged to an individual sequence type (Table 3). All *Y. pseudotuberculosis* O:1 strains carried the same virulence genes (*ail*, *inv*, *irp*, and *psa*A) located in the chromosome. Three (25%) strains were missing the *virF* and *yad*A genes located on the pYV plasmid (Table 3). Two sequence types (ST9 and ST42) were identified among the *Y. pseudotuberculosis* O:1 strains.

Most (92%, 35/38) *Y. enterocolitica* strains were susceptible to 13 out of 15 antimicrobials included in the EUVSEC3 kit. Only intrinsic resistance to ampicillin and streptogramin was observed (Table 4). Three strains expressed high MIC values to sulfamethoxazole, two of which carried the *sul*1 gene and one carried the *sul*2 gene. Two strains carried the *cat*A1 gene and were resistant to chloramphenicol, and one strain was resistant to tetracycline and carried the *tet*(A) gene. Additionally, three strains carried genes coding for streptomycin resistance (Table 4). The *Y. pseudotuberculosis* strains were susceptible to most (14/15) of the tested antimicrobials; only intrinsic resistance to colistin was observed. Three *Y. enterocolitica* 4/O:3 strains were multi-drug-resistant (MDR): two MDR strains showed resistance to chloramphenicol (*cat*A1), streptomycin (*aad*A12), and sulfamethoxazole (*sul*1), and one MDR strain showed resistance to streptomycin (*aph*), sulfamethoxazole (*sul*2), and tetracycline (*tet*A).

In total, 17 sequence types (CTs) based on cgMLST were identified among 26 *Y. enterocolitica* bioserotype 4/O:3/ST135 strains (Table 5). The ADs varied between 1 and 189. The *Y. enterocolitica* bioserotype 2/O:9/ST139 strains contained three CTs, which differed from each other by 2 to 37 ADs. All eight non-pathogenic *Y. enterocolitica* 1A strains differed from each other by more than 1600 ADs (Table 5). *Y. pseudotuberculosis* O:1/ST9 strains differed from each other by 20 to 93 Ads, and ST42 strains by 1 to 78 ADs.

Two clusters including two highly related strains in both clusters were observed among *Y. enterocolitica* 4/O:3/ST135 strains (Figure 2). Two MDR *Y. enterocolitica* 4/O:3/ST135 strains belonging to CT69 were highly related with one allelic mismatch, and these were found in dogs from the same litter. Additionally, two strains belonging to CT4483 showing only one AD were isolated from dog faeces sampled in 2019 and 2020. All other strains differed from each other by more than 10 ADs, which was the maximum number of ADs for a cluster definition [23]. Two strains with CT782 among the four bioserotype 2/O:9/ST139 strains formed a small, highly related cluster with only two ADs between the strains (Table 5). These strains were isolated in 2018 and 2021. The strains belonging to the non-pathogenic biotype 1A were very heterogenous compared to the pathogenic 4/O:3 and 2/O:9 strains.

## 4. Discussion

*Y. enterocolitica* of the bioserotype 4/O:3 was the most common yersinia found in dog faeces in our study. We detected the most important chromosomal virulence genes (*ail* and *yst*A) in all bioserotype 4/O:3 strains. The virulence plasmid (pYV), which can easily be lost during culturing, was missing in only one strain. This showed that the bioserotype 4/O:3 strains we identified in dogs and cats could be regarded as pathogenic, capable of causing yersinosis. However, canine yersiniosis has barely been investigated. In an earlier study, Fenwick et al. [24] demonstrated that asymptomatic dogs can excrete *Y. enterocolitica* 4/O:3 in their faeces for weeks. Stamm et al. [10] found *ail*-positive *Y. enterocolitica* most frequently in under-one-year-old dogs. They also reported bioserotype 4/O:3 to be the most common type in dog faeces, which was in accordance with our study. Bioserotype 4/O:3 has frequently been found in fattening pigs at slaughter, and raw or undercooked pork has been shown to be the main infection source of this type [6]. Raw pork has been demonstrated to be a possible infection source of yersiniosis in dogs and cats [9]. Unfortunately, no information about possible raw feeding was available in our study.

We found *Y. enterocolitica* bioserotype 2/O:9 in some dogs, which is the second most reported bioserotype in humans in the EU, including Finland [3]. All these strains could be regarded as pathogenic because they carried all the studied essential virulence genes, including genes located on the pYV plasmid. Stamm et al. [10] also reported bioserotype 2/O:9 to be the second most frequently identified type in dog faecal samples. Bioserotype 2/O:9 has sporadically been found in ruminants and wild animals in the EU, including Finland [25,26]; however, the infection sources and transmission routes remain unclear. Our study showed that dogs should be regarded as one source of bioserotype 2/O:9 infections among humans with close dog contact. Also, cats cannot be excluded, but the number of cat strains was very small in our study.

We also identified *Y. enterocolitica* biotype 1A in dogs and cats, which is regarded as a non-pathogenic type. In one dog faecal strain, we detected the *ail* gene, which is typically found only in pathogenic strains. However, all strains were *yst*A- and pYV-negative, which are important virulence factors needed for the pathogenesis of yersiniosis. Stamm et al. [10] also found biotype 1A in some dog faecal samples, and all strains were *ail-* and pYV-negative. The significance of 1A strains in diarrhoea remains unclear. Interestingly, we identified one biotype 1A strain from an abscess. Biotype 1A is widely distributed in the environment and has been isolated from various sources; thus, cross-contamination cannot be excluded. Biotype 1A strains have also been found in the faecal samples of humans with diarrhoea, especially in Finland [27]. These strains are speculated to have unknown virulence mechanisms and should therefore be regarded as potentially pathogenic. One discussed virulence gene found only in non-pathogenic 1A strains is *yst*B, which we detected in some of the 1A strains in our study. However, more research is needed concerning the pathogenicity of biotype 1A strains.

The *Y. pseudotuberculosis* O:1 strains were found in many faecal and extra-intestinal pet samples in our study. Serotype O:1 is the most common serotype reported in human and non-human samples worldwide [2]. Serotype O:1 has been identified in several foodborne outbreaks in Finland; however, the transmission routes remain unclear [28,29]. Wild animals, especially rodents and birds, are regarded as the most important reservoirs of *Y. pseudotuberculosis* [2]. Contaminated fresh produce and raw milk have been identified as the most frequent vehicles in human outbreaks. Outdoor cats can be infected, e.g., by eating birds and rats or by drinking raw milk. However, further studies are needed to clarify the possible infection sources and transmission routes for pets. In our study, *Y. pseudotuberculosis* O:1 was identified in most extra-intestinal samples.

We only identified ST135 among pathogenic *Y. enterocolitica* 4/O:3 strains and ST139 among pathogenic 2/O:9 strains using 7-gene MLST. These sequence types were heterogeneous, showing numerous ADs and several CTs with cgMLST. However, they were more homogeneous compared with the non-pathogenic 1A strains, which all belonged to individual STs and CTs with more than 1600 ADs. This high heterogeneity among the 1A strains could be explained by their large environmental distribution, which makes it challenging to interpret their true relevance in the pathogenesis of yersiniosis. We additionally identified two sequence types, ST9 and ST42, among pathogenic *Y. pseudotuberculosis* strains, which clearly differed from each other using cgMLST. Both sequence types have been reported in humans and animals in Finland in previous studies [28,29]. Only three small clusters were observed among the *Y. enterocolitica* strains. However, most *Y. enterocolitica* and *Y. pseudotuberculosis* strains differed by more than 10 ADs, which was the cut-off for a cluster. This indicated that the yersiniosis cases in our study were sporadic, with no common sources. This was expected because the number of strains was small, and the strains were collected over quite a long period. The next step is to design a prospective epidemiological study to compare pet strains with human and food strains using WGS and to collect comprehensive historical data from pets with yersinosis.

Nearly all the *Y. enterocolitica* strains were susceptible to most of the clinically important microbial agents. Intrinsic ampicillin resistance, which was observed among *Y. enterocolitica* strains, is linked to the *bla*A gene located in their chromosome [4]. Also, all the *Y. pseudotuberculosis* strains were very susceptible to most of the studied antimicrobials, and only high MIC values to colistin were observed. Colistin resistance is expected in *Y. pseudotuberculosis* without the colistin resistance gene and is most likely due to chromosomal mutations conferring the intrinsic resistance [30]. MDR *Y. enterocolitica* strains have rarely been found in Europe except in Spain and Italy [5]. We could identify two MDR *Y. enterocolitica* 4/O:3/ST135/CT69 strains from dogs that carried resistance genes against chloramphenicol (*cat*A1), streptomycin (*aad*A1), and sulfonamides (*sul*1). These genes have recently been reported in *Y. enterocolitica* 4/O:3 outbreak strains in Sweden [31]. The two strains in our study were highly related, and they originated from two dogs from the same litter, indicating a common infection source. Unfortunately, this was the only historical information we have, and raw feeding could not be excluded. The third MDR strain (*Y. enterocolitica* 4/O:3/ST135/CT1010) was resistant to sulfonamides (*sul*2), streptomycin (*aph*3-ib and *aph*6-id), and tetracycline (*tet*A), which are antimicrobials commonly used to treat animals.

Most (84%) of the *Yersinia* strains originated from dog faeces, and only three (6%) strains were from cat faeces. One reason may have been that the cat faecal samples were more rarely investigated than dog samples or that cats more rarely suffer from bacterial diarrhoea. We know that raw feeding of pets is a risk factor for developing diarrhoea and that *Y. enterocolitica* 4/O:3 is associated with raw pork [9,32]. We also know that young dogs are more prone to diarrhoea caused by *Y. enterocolitica*. To obtain more information about the incidence of enteric yersinosis in pets and to investigate any possible associations between animal age, feeding, and yersiniosis in dogs and cats, a prospective study should be conducted.

## 5. Conclusions

We showed that enteropathogenic *Yersinia* could be present in clinical faecal samples of dogs and cats. Therefore, yersiniosis should be included in the differential diagnosis of enteritis in pets. Yersiniosis should be kept in mind, especially when diarrhoea occurs in pets with a raw meat diet history. The most frequently identified *Y. enterocolitica* type was bioserotype 4/O:3, which is also the most common type among human yersiniosis. A possible link between human and pet yersiniosis needs to be studied further using a one-health concept with close cooperation between veterinarians and physicians. According to the WGS analyses used, most of the *Yersinia*-positive cases in our study were sporadic without a clear relationship with each other. Nearly all the *Yersinia* strains were susceptible to most of the clinically important microbial agents; only three MDR *Y. enterocolitica* strains were found.

## Figures and Tables

**Figure 1 pathogens-13-00054-f001:**
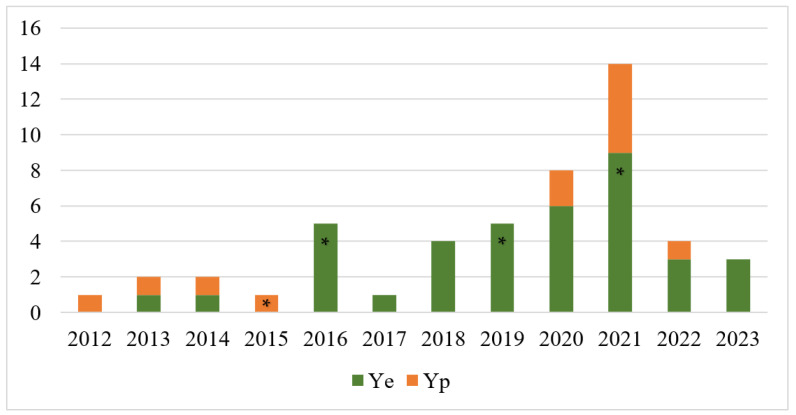
Number of *Y. enterocolitica* (Ye) and *Y. pseudotuberculosis* (Yp) strains from pets isolated between 2012 and 2023 in Finland. * Distribution of four cat strains.

**Figure 2 pathogens-13-00054-f002:**
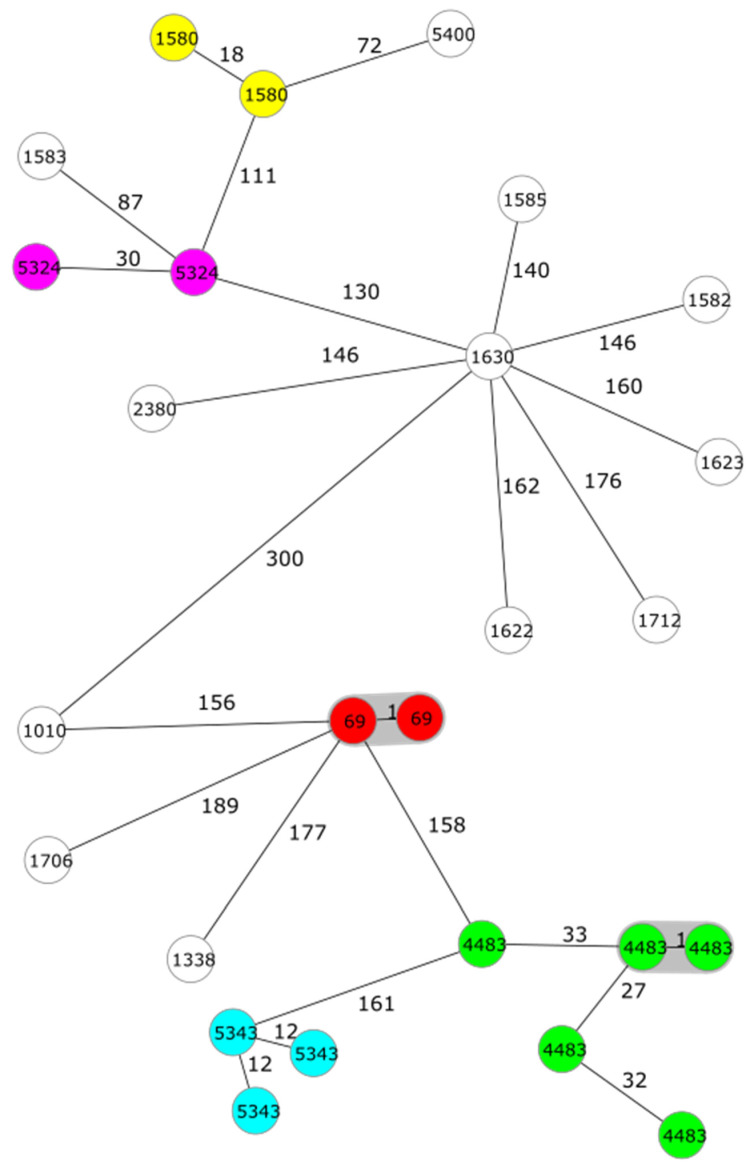
Minimum spanning tree (MST) of 26 *Yersinia enterocolitica* 4/O:3 strains based on cgMLST (3839 targets). Number of allelic differences (ADs) between the strains are indicated on the connecting lines. Nodes are numbered with sequence types (CTs) according to Table 5. Clusters are shaded in grey and a cluster distance threshold of a maximum of 10 ADs was used. CTs including more than one strain are marked with different colours.

**Table 1 pathogens-13-00054-t001:** Target genes used for identification and characterisation of *Y. enterocolitica* (Ye) and *Y. pseudotuberculosis* (Yp) strains.

Identification	Target	Gene	Ye	Yp	Reference
Species	16SrRNA	*rrn*	x	x	[12]
		*rrn*	x		[13]
Pathogenicity	Chromosome	*inv*	x		[14]
		*inv*		x	[15]
		*ail*	x		[16]
		*ail*		x	[17]
		*yst*A	x		[18]
		*yst*B	x		[18]
	Plasmid	*yad*A	x	x	[19]
		*vir*F			[20]
Serotype	O:3	*rfb*C			[21]
	O:9	*per*			[22]

**Table 2 pathogens-13-00054-t002:** Phenotypic characteristics of 38 *Y. enterocolitica* (Ye) and 12 *Y. pseudotuberculosis* (Yp) strains from dogs and cats.

Animal	Number of Strains	Sample Type	Ye		Yp
			Biotype	Serotype	Serotype
Dog	6	Faeces (5), abscess (1)	1A	NT ^a^	
	4	Faeces	2	O:9	
	25	Faeces	4	O:3	
	11	Faeces (7), urine (2), abscess (1), blood (1)			O:1
Cat	2	Faeces	1A	NT	
	1	Faeces	4	O:3	
	1	Wound			O:1

^a^ not typed.

**Table 3 pathogens-13-00054-t003:** Virulence genes present in 38 *Y. enterocolitica* (Ye) and 12 *Y. pseudotuberculosis* (Yp) strains belonging to different sequence types.

Sp.	Type	No. of Strains	Virulence Gene							Sequence Type ^a^
			*ail*	*inv*	*irp*	*psa*A	*yst*A	*yst*B	*vir*F/yadA	
Ye	1A	1	1	0	0	0	0	1	0	388
		1	0	0	0	1	0	0	0	138
		2	0	2	0	0	0	0	0	290, 738
		4	0	4	0	0	0	4	0	148, 179, 365, 479/551
	2/O:9	4	4	4	0	4	4	0	4	139
	4/O:3	26	26	26	0	26	26	0	25	135
Yp	O:1	3	3	3	3	3	0	0	2	9
		9	9	9	9	9	0	0	7	42

^a^ Using 7-gene multi-locus sequence typing (MLST) based on the Achtman scheme.

**Table 4 pathogens-13-00054-t004:** The minimum inhibitory concentration (MIC) values and antimicrobial resistance (AMR) genes among *Yersinia* strains isolated from dogs and cats.

Antimicrobial Agent	Number of Strains	MIC (µg/mL)		AMR Gene Present
		Breakpoint S≤	Observed Value	
*Y. enterocolitica* (n = 38)				
Ampicillin	38	8	≥32 ^a^	*bla*A
Chloramphenicol	2	8	>64	*cat*A1
Streptomycin	2		NT ^b^	*aad*A12
	1		NT	*aph*(3)-lb, *aph*(6)-ld
Streptogramin	38		NT	*vat*(F)
Sulfamethoxazole	2	NA ^c^	>512	*sul*1
	1	NA	>512	*sul*2
Tetracycline	1	4	>32	*tet*(A)
*Y. pseudotuberculosis* (n = 12)				
Colistin	12	2	>16 ^a^	0

^a^ Intrinsic resistance; ^b^ not tested; ^c^ not available.

**Table 5 pathogens-13-00054-t005:** Sequence types and allelic differences (ADs) of *Y. enterocolitica* (Ye) and *Y. pseudotuberculosis* (Yp) strains based on 7-gene multi-locus sequence typing (MLST) (ST) and core genome MLST (CT).

Species	Type	ST	CTs (Number of Strains)	ADs
Ye	4/O:3	135	69 (2), 1580 (2), 4483 (5), 5324 (2), 5343 (3)	1–50
			1583, 5400	51–100
			1010, 1338, 1582, 1585, 1622, 1623, 1630, 1706, 1712, 2380	101–189
	2/O:9	139	510 (2), 904, 5351	2–37
	1A	138	5575	2025
		148	1401	1661
		179	1295	1661
		290	1379	2030
		365	2282	2067
		388	1710	2164
		479/551	5505	2650
		738	5332	2025
Yp	O:1	9	782 (2), 806	20–93
		42	816, 1728, 3900 (2), 5560 (3), 5610 (2)	1–78

## Data Availability

The data analysed during the study are available from the corresponding author on reasonable request.

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
