# Peer review of "Enteropathogenic Yersinia with Public Health Relevance Found in Dogs and Cats in Finland"

_pathogens, 2024, doi:10.3390/pathogens13010054_

Round 1
Reviewer 1 Report
Comments and Suggestions for Authors
Brief summary: This manuscript describes the characterization of enteropathogenic Yersinia isolated in dogs and cats in Finland. The authors show that the species and subspecies found in these pets are similar to those found in humans and suggest that dogs and cats could serve as an infection source for human yersiniosis.
General comments: This study is very interesting as few studies of enteric yersiniosis in dogs and cats are available in the litterature. The genotypic characterization of the Yersinia strains confirms the phenotypic characteristics and identifies the virulence and resistance genes of the strains.
Associated data about the clinical condition of the animals would have been interesting to compare the symptoms caused by pathogenic and non-pathogenic Yersinia.
Specific comments:
Materials and Methods
Ligne 76: animals instead of patients.
Line 86: Y. enterocolitica strains were biotyped based on tween esterase activity, indole production and xylose and trehalose fermentation. These tests seem to be part of the biotyping scheme of Wauters. Did you also perform the test for the pyrazinamidase activity? How could you distinguish biotype 1A from biotype 1B?
Results
Line 182: Please explain how was determined the threshold (10 ADs) used for the cluster definition with the 3839 genes-cgMLST specific to Y. enterocolitica or if it is simpler cite a reference in Materials and Methods.
Author Response
“patients” has been changed to “animals” (line 77).
No, we did not perform PYZ. We distinguished high pathogenic biotype 1B strains from non-pathogenic 1A strains by screening the virulence genes.
A reference [23] has been added into the text (line 187).
Reviewer 2 Report
Comments and Suggestions for Authors
Dear authors, the comments and recommendations are included in the attached file

Author Response
“Epidemioloy” has been added into the text (lines 14 and 52).
“One health concept” has now been mentioned in the conclusions (lines 297-298).
“Clinical samples” has been added into the last paragraph pf the introduction (line 62).
Repetition in the abstract was difficult to find; however, we deleted some words (lines 17-18 and 24-25).
We added “bacterial” before diarrhoea (line 283).
The text marked bold in the conclusions part has been unified and modified according to the reviewer’s suggestion (lines 292-294).
AMR results have now been mentioned in the conclusion part (lines 300-302).